# The Presence of Hyperhomocysteinemia Does Not Aggravate the Cardiometabolic Risk Imposed by Hyperuricemia in Young Individuals: A Retrospective Analysis of a Cross-Sectional Study

**DOI:** 10.3390/ijerph192013521

**Published:** 2022-10-19

**Authors:** Katarína Šebeková, Radana Gurecká, Gabriela Repiská, Ivana Koborová, Ľudmila Podracká

**Affiliations:** 1Institute of Molecular Biomedicine, Faculty of Medicine, Comenius University, 811 08 Bratislava, Slovakia; 2Institute of Medical Physics, Biophysics, Informatics and Telemedicine, Faculty of Medicine, Comenius University, 813 72 Bratislava, Slovakia; 3Institute of Physiology, Faculty of Medicine, Comenius University, 813 72 Bratislava, Slovakia; 4Departemnt of Pediatrics of the Faculty of Medicine, Comenius University and The National Institute of Children’s Health, 833 40 Bratislava, Slovakia

**Keywords:** uric acid, homocysteine, adolescents, cardiometabolic risk factors, continuous metabolic syndrome score, ADMA

## Abstract

Background: Little research has been conducted into the effects of the combined manifestation of hyperuricemia and hyperhomocysteinemia on cardiometabolic risk factors and markers in young subjects. Methods: 1298 males and 1402 females, 14-to-20-year-olds, were classified into four groups: 1/normouricemic/normohomocysteinemic, 2/normouricemic/hyperhormohomocysteinemic, 3/hyperuricemic/normohomocysteinemic, and 4/hyperuricemic/hyperhomocysteinemic. Anthropometric measures, blood pressure, plasma glucose, insulin, lipids, markers of renal function, C-reactive protein, asymmetric dimethylarginine, and blood counts were determined. Results: Hyperuricemic males (but not females) had higher odds for hyperhomocysteinemia than normouricemic ones (OR: 1.8; 95% CI: 1.4–2.3; *p* < 0.001). Homocysteine and uric acid levels correlated directly (males: r = 0.076, females: r = 0.120; *p* < 0.01, both). Two-factor analysis of variance did not reveal a significant impact of hyperhomocysteinemia on any of the investigated cardiometabolic variables in females; in males, hyperuricemia and hyperhomocysteinemia showed a synergic effect on asymmetric dimethylarginine levels. Among four groups, subjects concurrently manifesting hyperuricemia and hyperhomocysteinemia did not presented the highest continuous metabolic syndrome score—a proxy measure of cardiometabolic risk; neither the multivariate regression model indicated a concurrent significant effect of uric acid and homocysteine on continuous metabolic syndrome score in either sex. Conclusion: In young healthy subjects, hyperhomocysteinemia does not aggravate the negative health effects imposed by hyperuricemia.

## 1. Introduction

Uric acid (UA) is a bioactive end-product of the metabolism of purines (adenine and guanine). Extracellular UA acts as an antioxidant; intracellularly, UA exerts pro-oxidant effects [1]. Experimental studies document that UA affects vascular cell functions via promotion of degradation of the vasodilator nitric oxide and boosting of activity of the renin-angiotensin axis [2]. Both UA and xanthine oxidoreductase (the enzyme that generates uric acid) may induce oxidative stress, promote inflammation, endothelial dysfunction, and atherosclerosis [2,3]. Hypertension, kidney, and cardiovascular diseases, fatty liver, dyslipidemia, obesity, insulin resistance, metabolic syndrome (MetSy), or diabetes are associated with elevated serum UA (SUA) levels already in adolescents [4,5,6].

Homocysteine (Hcy) is a sulfur-containing amino acid produced from the essential amino acid methionine as a byproduct of trans-methylation reactions. It derives from *S*-adenosyl-homocysteine hydrolase-catalyzed hydrolysis of *S*-adenosylhomocysteine to adenosine. Experimental studies show that at the molecular level, the toxicity of Hcy includes mechanisms involving the formation of reactive oxygen species, hypomethylation, induction of unfolded protein response, and protein homocysteinylation. At the cellular level, Hcy is a pro-inflammatory, pro-thrombotic, pro-atherogenic factor, vasodilation impairing agent, and an inducer of endoplasmatic reticulum stress [7]. Even in children and adolescents, increased levels of Hcy are associated with a range of disorders, such as renal and cardiovascular diseases, obesity, diabetes, premature atherosclerosis, or impaired bone health [8,9,10,11].

Uric acid and Hcy are metabolically interrelated: adenosine produced during metabolic transit of Hcy might eventually be metabolized into UA. Most daily UA and Hcy disposal occur via kidneys. Both UA and Hcy may affect the oxidative status, exert proinflammatory and proatherogenic effects, and may alter vascular cell function via interfering with NO metabolism. Levels of SUA and Hcy rise in identical pathologies, and both variables display a positive direct relationship even in healthy adults [12,13,14,15,16,17]. A synergistic association of hyperuricemia and hyperhomocysteinemia (hyperHcy) with chronic kidney disease has been documented in middle-aged and elderly patients [18]; while combined hyperuricemia and hyperHcy additively increased the risk of manifestation of subclinical atrial fibrillation in patients with cardiac implantable electronic devices [19]. However, it is not fully clarified whether hyperuricemia and hyperHcy represent markers, or rather act as etiological agents in cardiovascular and other mentioned pathologies [20,21]. The current guidelines of professional societies do not consider SUA or Hcy as cardiovascular disease risk stratifiers.

The recent cross-sectional analysis indicated that in adolescents, Hcy levels were positively correlated with those of SUA, and the odds to present elevated SUA levels increased across the Hcy terciles [22]. Yet, it remains unclear whether the combined manifestation of elevated SUA levels and hyperHcy exerts additive effects on cardiometabolic risk factors and markers in young healthy subjects. This question is of particular importance, as there is robust evidence linking cardiovascular risk factors in childhood and adolescence (even in juveniles with mildly elevated risk factor scores) with clinical atherosclerotic cardiovascular disease events in adulthood [23].

We hypothesized that concurrent manifestation of elevated SUA levels and hyperHcy exerts an additive worsening effect on cardiometabolic risk factors or markers compared with the presence of isolated hyperuricemia or hyperHcy. To this point, we retrospectively analyzed data obtained from healthy young individuals.

## 2. Materials and Methods

### 2.1. Subjects

The cross-sectional “Respect for Health” study has been described previously [22]. Briefly, students of state secondary schools in the Bratislava Region participated voluntarily in the survey. Exclusion criteria were any acute or chronic illness, pregnancy, or lactation in females.

Anthropometric, blood chemistry, and hematology data were obtained from 2960 students aged 11-to-23 years. We excluded 260 subjects (9 non-Caucasians (4 Vietnamese, 3 Koreans, 2 others; due to unavailable reference values of blood chemistry variables for minorities), 6 potential diabetics (fasting plasma glucose (FPG) > 6.9 mmol/L), aged < 14 (*n* = 11) or >20 (*n* = 14) years, 220 with missing values), leaving 2700 individuals (51.9% females) aged 14-to-20 years for the present analysis.

The study was approved by The Ethics Committee of the Bratislava Self-governing Region and conformed to the Helsinki Declaration. In minors, participation was subject to the written informed consent of the legal representative and the verbal assent of the child. Written informed consent was obtained from full-aged participants.

### 2.2. Measurements

Anthropometric measurements were performed following standard guidelines, as described previously [24]. Briefly, height was measured using a portable extendable stadiometer, waist circumference using a flexible tape, and body weight employing digital scales (Omron BF510, Kyoto, Japan). BMI and waist-to-height ratio (WHtR) were calculated.

Blood pressure (BP) was measured on a dominant arm in a person relaxed for at least 5 min in the seated position, using a digital monitor (Omron M-6 Comfort, Kyoto, Japan). The mean of the last two measurements was recorded.

After overnight fasting, venous blood and spot urine were collected. In the central laboratory, serum chemistry (glucose, triacylglycerols (TAG), high-density lipoprotein cholesterol (HDL-C), UA, insulin, creatinine, high-sensitivity C-reactive protein (CRP)) and urine (creatinine, albumin) analyses (Advia 2400 analyzer, Siemens, Germany) and blood counts (Sysmex XE-2100 analyzer, Sysmex Corporation, Kobe, Japan) were performed using standard laboratory methods, as described previously [24]. Total plasma L-homocysteine was measured with a fluorescence polarization immuno-812 assay (IMX; Abbott Diagnostics, Maidenhead, Berkshire, UK). Asymmetric dimethylarginine (ADMA) was determined at the Institute of Molecular Biomedicine, using commercial ELISA kits (DLD Diagnostika GmbH, Hamburg, Germany) according to the manufacturer’s instructions. Low-density lipoprotein cholesterol (LDL-C) was calculated employing the Friedewald formula; insulin sensitivity using the Quantitative Insulin Sensitivity Check Index (QUICKI) [25]; the atherogenic index of plasma (AIP) as log(TAG)/HDL-C) [26]; and estimated glomerular filtration rate (eGRF) employing the equation for the full-age spectrum with Q-height extension [27]. Cardiometabolic risk was estimated using the continuous metabolic syndrome score as WHtR/0.5 + fasting glycemia/5.6 + SBP/130 + TAG/1.7 − HDL-C/1.02 (males) or 1.28 (females) [28]. The urinary albumin-to-creatinine ratio was calculated.

### 2.3. Definition of Elevated Uric Acid Levels, Hyperuricemia, Hyperhomocysteinemia, Cardiometabolic Risk Factors, and Metabolic Syndrome

Employing reference ranges of the laboratory of the National Institute for Children’s Diseases in Bratislava, we classified hyperuricemia as SUA concentration >340 μmol/L in females; in males >360 μmol/L if aged < 17 years, and >420 μmol/L in those aged ≥ 17 years; individuals aged 15 and 16 years who presented with Hcy > 10.0 μmol/L, adolescents aged 16 and 17 years who displayed Hcy > 11.3 μmol/L, and those aged ≥ 18 years with Hcy concentrations >15.0 μmol/L were classified as hyperhomocysteinemic.

In 14-to-17-year-olds, general overweight/obesity was classified according to the international age- and sex-specific cutoff points for BMI [29], in individuals aged ≥ 18 years as BMI ≥ 25 kg/m^2^. Central obesity was defined as WHtR ≥ 0.5 [30]. The increased cardiometabolic risk was classified according to guidelines for the classification of metabolic syndrome components: systolic BP ≥ 130 mmHg, DBP ≥ 85 mmHg, triacylglycerols ≥ 1.7 mmol/L, fasting glycemia ≥5.6 mmol/L, and HDL-C as <1.03 mmol/L in males and females aged < 16 years; and <1.29 mmol/L in females aged ≥ 16 years [31]. Subjects presenting at least three cardiometabolic risk factors, e.g., central obesity, elevated fasting glycemia, BP, TAG, or low HDL-C concentrations were considered as suffering from metabolic syndrome [31]. Moreover, fasting insulin ≥20 μIU/mL [32], CRP > 3 mg/L [33], atherogenic index ≥0.11 [26], and the presence of microalbuminuria/albuminuria (urinary albumin/creatinine ≥2.5 mg/mmol in males and ≥3.5 mg/mmol in females [24]) were considered as markers of increased cardiometabolic risk. The number of cardiometabolic risk factors (RF) was calculated as a sum of the presence of binary coded elevated BP, adiposity (presence of central obesity or general overweight/obesity), dyslipidemia (elevated TAG or AIP or low HDL-C), alteration of glucose metabolism (elevated fasting glycemia or insulinemia), and elevated CRP.

### 2.4. Statistical Analyses

Data not fitting the normal distribution (Shapiro–Wilk test) were log-transformed before statistical analyses. Males and females were compared using the two-sided independent samples Student’s *t*-test. According to the reference ranges of SUA and Hcy, subjects were classified into four groups: 1/concentrations of SUA and Hcy below the upper reference range (e.g., normal, *n*), 2/SUA levels below and Hcy above the upper reference range, 3/SUA above and Hcy concentration below, and 4/both SUA and Hcy concentrations above the reference ranges. Four groups were compared using the two-factor analysis of variance (ANOVA) with the presence/absence of hyperuricemia and presence/absence of hyperHcy as fixed factors. Normally distributed data are given as mean ± standard deviation (SD), those failing assumptions of normality are described with a back-transformed geometric mean (interval −1 SD, +1 SD). Categorical data were compared using the Fisher’s exact test or Chi-square test with Yates´ correction (Y) if appropriate, and are given as counts and frequencies. Pearson correlation coefficients and odds ratios (OR) were calculated. *p* value < 0.05 was considered significant. Statistical software SPSS version 16 (SPSS, Chicago, IL, USA) was used.

Multivariate regression of independent factors on continuous metabolic syndrome score was performed using the orthogonal projection to latent structures model (OPLS, Simca v.16 software, Sartorius Stedim Data Analytics AB, Umea, Sweden). In Model 1, age, insulin, CRP, eGFR, ACR, leukocyte (WBC) and erythrocyte (RBC) counts, and ADMA, were entered as independent variables; in Model 2, SUA was forced into the model; in Model 3 SUA was replaced by Hcy; in Model 4 both SUA and Hcy were entered. Before fitting the OPLS models, all variables with high skewness and a low min/max ratio were log-transformed and all data were mean-centered. Variables with a variable of importance for the projection (VIP) values ≥ 1.00 were considered significant.

## 3. Results

Cohort characteristics are given in Table 1. Males differed from females in all variables except for age, QUICKI, the prevalence of elevated fasting insulinemia, elevated TAG levels, or microalbuminuria/albuminuria.

### 3.1. Males

SUA and Hcy levels ranged 142–563 µmol/L, and 1.7–63.3 µmol/L, respectively. There was a significant direct relationship between lnHcy and SUA levels (r = 0.076, *p* = 0.006).

#### 3.1.1. Correlations between Cardiometabolic Risk Factors and Markers with Uricemia or Homocysteinemia

Correlations between age or glycemia and SUA were insignificant; significant inverse correlations were revealed between SUA and HDL-C, QUICKI, ln urinary albumin/creatinine, and eGFR; all other variables showed a direct significant relationship with SUA (Table 2). Age, DBP, non-HDL-C, AIP, and lnTAG showed a positive significant relationship with lnHcy; while eGFR, ln urinary albumin/creatinine, and lnADMA correlated inversely. However, all significant associations were weak (Pearson r: SUA: −0.062-to−0.316, lnHcy: −0.056-to−0.094; Table 2).

#### 3.1.2. The Effects of Uricemia and Homocysteinemia on Cardiometabolic Variables

30.9% of males suffered from hyperuricemia, 32.6% displayed hyperHcy; 12.8% manifested combined hyperuricemia and hyperHcy (Table 3). Among 423 hyperhomocysteinemic males, 34 displayed moderate hyperHcy (>30 μmol/L), and 9 out of 34 suffered from hyperuricemia. Compared with males displaying SUA levels within the reference range, those with hyperuricemia had increased odds for hyperHcy (OR: 1.76 (95% confidence interval [CI]: 1.38–2.49; *p* < 0.001).

To investigate whether the concurrent manifestation of hyperuricemia and hyperHcy is associated with worsening of cardiometabolic variables, 2-factor ANOVA was performed. Hyperuricemia was associated with higher BMI, WHtR, systolic and diastolic BP (SBP, DBP), fasting insulinemia, non-HDL-C concentrations, lnTAG, lnCRP, lnADMA, atherogenic index of plasma, continuous metabolic syndrome score, risk factors number, and erythrocyte counts; lower insulin sensitivity and HDL-C levels. HyperHcy independently affected 6 variables, and there was a concurrent independent opposite effect of hyperuricemia and hyperHcy on BMI, WHtR, SBP, number of cardiometabolic risk factors, or lnCRP, and synergic one on lnADMA (Table 3).

The prevalence of MetSy was the highest in males manifesting isolated hyperuricemia (nSUA/nHcy: 2.5%, nSUA/hyperHcy: 1.6%, hyperuricemia/nHCY: 8.5%, hyperuricemia/hyperHcy: 7.8%; *p* < 0.001).

#### 3.1.3. Multivariate Regression Models

The OPLS multivariate regression model was employed to elucidate whether and how SUA and/or Hcy affect the continuous metabolic syndrome score—a proxy measure of cardiometabolic risk. If neither SUA nor Hcy was considered, the model indicated that insulinemia and inflammatory markers independently affect the continuous metabolic syndrome score (Table 4, Model 1). After the inclusion of SUA, it became an additional significant predictor of cardiometabolic risk, regardless of the absence (Model 2) or presence (Model 4) of Hcy in the model. However, the variance of the continuous metabolic syndrome score explained by models 2 and 4 increased only slightly (by 3%) after the inclusion of SUA. After forcing Hcy into the model, erythrocyte counts but not Hcy became a significant predictor of continuous metabolic syndrome score (Models 3 and 4). The variability of the continuous metabolic syndrome score explained by the models was not affected.

### 3.2. Females

Serum uric acid levels ranged 123–578 µmol/L, those of homocysteine 2.0–37.7 µmol/L. SUA and lnHcy showed a direct significant correlation (r = 0.120. *p* < 0.001).

#### 3.2.1. Correlations between Cardiometabolic Risk Factors and Markers with Uricemia or Homocysteinemia

BMI, WHtR, continuous metabolic syndrome score, risk factors number, erythrocytes count, lnADMA, lnCRP, SBP, DBP, leukocytes count, atherogenic index of plasma, and non-HDL-C displayed a positive, while eGFR, HDL-C, age, and ln urinary albumin/creatinine ratio showed a negative significant relationship with SUA (Table 2). LnHcy positively correlated with age, lnTAG, continuous metabolic syndrome score, atherogenic index of plasma, leukocyte counts, and non-HDL-C; and inversely with eGFR and lnADMA. Similar to males, all significant correlations were weak (Pearson r: SUA: −0.043-to−0.253, lnHcy: −0.053-to−0.131; Table 2).

#### 3.2.2. The Effects of Uricemia and Homocysteinemia on Cardiometabolic Variables

The prevalence of hyperuricemia reached 5.8%, that of hyperHcy was 14.9%, and both markers were concurrently elevated in 1.1% of females (Table 5). Among 209 hyperhomocysteinemic females, 5 displayed moderate hyperHcy (>30 µmol/L); 1 of them suffered from hyperuricemia. Compared with the normouricemic females, those presenting with hyperuricemia did not have increased odds to manifest hyperHcy (OR: 1.32; 95% CI: 0.74–2.34; *p* = 0.349).

Two-factor ANOVA indicated that hyperuricemia was associated with higher BMI, WHtR, TBF, SBP, DBP, AIP, continuous metabolic syndrome score, risk factor number, lnCRP, and leukocyte count; and lower HDL-C and eGFR (Table 5). The model neither indicated a significant impact of hyperHcy on either variable nor a significant SUA/hyperHcy interaction. No female displaying concurrently hyperuricemia and hyperHcy manifested MetSy (nSUA/nHcy: 1.2%, nSUA/hyperHcy: 1.0%, hyperuricemia/nHCY: 6.1%, hyperuricemia/hyperHcy: 0%; *p*
**=** 0.039^Y^).

#### 3.2.3. Multivariate Regression Models

The multiple regression model not adjusted for SUA and Hcy selected insulinemia, inflammatory markers, and ADMA as significant independent predictors of continuous metabolic syndrome score (Model 1; Table 4). Forcing SUA (Model 2), Hcy (Model 3), or their combination (Model 4) into the model neither affected the selection of independent variables modulating the continuous metabolic syndrome score, nor its variability explained by the OPLS model, which was generally low.

## 4. Discussion

Retrospectively, we tested the hypothesis that in healthy young individuals, the concurrent presence of hyperuricemia and hyperHcy is associated with less favorable cardiometabolic status in comparison with the manifestation of only one of them. We did not confirm our hypothesis. In males, two-factor ANOVA indicated that out of nineteen investigated risk factors and markers, hyperuricemia significantly affected fifteen; hyperHcy only five. Paradoxically, hyperHcy was associated with lower measures of obesity, lower SBP, number of manifested risk factors, and CPR concentration regardless of the absence or presence of hyperuricemia. The synergic effect of hyperuricemia and hyperHcy was observed only for lnADMA. In females, hyperuricemia was associated with worsening of eleven cardiometabolic risk factors and markers; while none of the endpoints was affected significantly by hyperHcy. No significant interaction between hyperuricemia and hyperHcy was observed in either sex.

Numerous studies in the general population of adults indicate that SUA and Hcy levels show a linear positive correlation [12,13,14,15,17]. In line with the recent study on teenagers [22], we show that this direct relationship is manifested already in young healthy subjects—a population not affected by age-associated comorbidities. Similar to other studies [14,19,34], this correlation was tighter in females compared with males, despite that females generally present with lower levels of SUA and Hcy, and a lower prevalence of cardiometabolic risk factors and markers compared with males. As in the aforementioned studies, correlations between SUA and Hcy were weak. This evokes a question of whether such weak statistical correlations might be of clinical impact.

Associations between uricemia or homocysteinemia and variables characterizing cardiometabolic risk are widely studied in different populations. In the general population of adolescents, rising SUA concentrations go hand in hand with worsening of the components of MetSy, other cardiovascular disease indicators, such as inflammatory markers or glomerular filtration rate, as well as increased cardiometabolic risk [35,36,37,38]. With some minor sex differences, several cardiometabolic risk factors and markers worsened, and the number of cardiometabolic risk factors and cardiometabolic risk (evaluated as continuous metabolic syndrome score) increased significantly with increasing SUA levels also in our study.

The reports on whether variables characterizing cardiometabolic status worsen with increasing Hcy levels in adults are inconsistent [12,13,14,15,16,17]. Large studies in adolescents show that neither MetSy nor the rising number of its components is associated with increased Hcy levels [39,40]. In our study, all significant simple correlations indicated a less favorable cardiometabolic status with increasing lnHcy in both sexes, albeit significant correlations were less frequent compared with those observed for SUA. However, two-factor ANOVA indicated a concurrent significant impact of hyperuricemia and hyperHcy only in males; and in five out of six cases—BMI, WHtR, SBP, risk factors number, and lnCRP—the presence of hyperHcy paradoxically associated with a partial amelioration of the negative effect of hyperuricemia. The cross-sectional nature of our study does not allow for commenting on potential mechanisms. We are not aware of similar reports in the literature; thus, our findings open the field for further research.

Although the combination of hyperuricemia and hyperHcy was associated with the highest ADMA levels among the four groups, the effect was significant only in males; and was synergic, not additive. Reports on the association of SUA and ADMA in the general population are scarce. The Polish study reported elevated ADMA levels in adolescents with hyperuricemia [41]. The hyperhomocysteinemia-associated rise in ADMA might reflect the interconnection of their metabolism. S-adenosylmethionine methyltransferases and protein-arginine methyl transferases participate simultaneously in Hcy and ADMA synthesis and hyperHcy may increase ADMA levels by reducing the activity of dimethylarginine dimethylaminohydrolase—an enzyme that metabolizes ADMA [42,43]. Moreover, ADMA is eliminated also by renal excretion. Thus, some of the deleterious effects of hyperHcy may involve ADMA-related cardiovascular effects. ADMA acts as a competitive inhibitor of NO synthase and may cause a further decrease in the bioavailability of NO by increasing the production of reactive oxygen species [44]. Elevated serum ADMA is associated with MetSy, endothelial dysfunction, and cardiovascular diseases such as hypertension and atherosclerosis [45].

In our adolescents, the prevalence of combined hyperuricemia and hyperHcy was low, particularly in females. Thus, the study could be underpowered concerning the ability of two-factor ANOVA to detect an additive effect of both factors. To this point, we used multivariate regression to test whether and how uricemia and/or homocysteinemia affect the continuous metabolic syndrome score. However, only SUA appeared as a significant independent predictor, and only in males; and the addition of both biomarkers into baseline prediction models did not improve their prediction abilities for the continuous metabolic syndrome score in either sex.

There are several limitations associated with this study. This is a retrospective analysis of a cross-sectional study; thus, a causal relationship cannot be inferred. The measurements were taken at a one-time point. Our results cannot be generalized to different populations. Information on circulating levels of vitamins essential for Hcy metabolism, genetics, lifestyle factors, or dietary habits that could potentially affect SUA or Hcy levels were not available. However, the study in Slovak adolescents indicated that 677 C → T mutation of the methylenetetrahydrofolate reductase gene, one of the most frequent genetic causes of moderate hyperHcy, was not associated with increased Hcy levels [46]. Regarding diets, daily intake of vitamin B12 and folate in these adolescents exceeded the recommended daily allowance, while that of vitamin B6 reached only 60–66%, thus could contribute to hyperHcy. On the other hand, to our knowledge, this is the largest study exploring the combined effects of hyperuricemia and hyperHcy on several cardiometabolic risk factors and markers in apparently healthy young individuals. The number of participants allowed for a separate evaluation of both sexes thus pointing out physiological sex disparities which may remain undetected if the sexes are not analyzed apart.

## 5. Conclusions

In our young healthy subjects, presence of combined hyperuricemia and hyperHcy was not associated with worse cardiometabolic status compared with that imposed by isolated hyperuricemia. While speculative, there might be several potential explanations. First, the relationship of SUA and Hcy with cardiometabolic risk variables rather reflects their metabolic interconnection than the pathophysiological link. It is also possible that in young healthy individuals SUA or Hcy may not be as specific as the conventional cardiometabolic risk factors concerning cardiovascular risk prediction. Our data do not rule out that there are (even non-clinical) populations where the combined hyperuricemia and hyperHcy associate with an increased cardiometabolic risk—thus, it requires validation in diverse cohorts. If combined risk is confirmed in certain populations, it remains to be elucidated whether successful intervention restoring SUA and Hcy to normal levels concomitantly reduces cardiometabolic risk. Moreover, potential sex-specific disease risk in later life imposed by combined hyperuricemia and hyperHcy should be evaluated.

## Figures and Tables

**Table 1 ijerph-19-13521-t001:** Cohort characteristics.

	Males	Females	*p*
N (%)	1298 (48.1)	1402 (51.9)	--
Age, years	17.2 ± 1.4	17.2 ± 1.4	0.274
Uric acid, µmol/L	354 ± 60	258 ± 51	**<0.001**
Homocysteine, μmol/mL	11.2 (7.7; 16.2)	9.4 (7.0; 12.7)	**<0.001**
Waist/height	0.44 ± 0.05	0.43 ± 0.05	**<0.001**
Body mass index, kg/m^2^	23.0 ± 3.9	21.9 ± 3.5	**<0.001**
Systolic blood pressure, mm Hg	122 ± 12	107 ± 9	**<0.001**
Diastolic blood pressure, mm Hg	73 ± 8	70 ± 8	**<0.001**
Glucose, mmol/L	4.9 ± 0.4	4.7 ± 0.4	**<0.001**
Insulin, μIU/mL	9.6 (5.7; 16.1)	10.0 (6.3; 16.0)	**0.013**
QUICKI	0.343 ± 0.027	0.343 ± 0.026	0.881
HDL-C, mmol/L	1.25 ± 0.23	1.52 ± 0.30	**<0.001**
Non-HDL-C, mmol/L	2.56 ± 0.68	2.74 ± 0.69	**<0.001**
Triacylglycerols, mmol/L	0.79 (0.51; 1.21)	0.79 (0.52; 1.19)	**0.047**
Atherogenic index	−0.19 ± 0.23	−0.26 ± 0.20	**<0.001**
cMSS	1.99 ± 0.48	1.83 ± 0.41	**<0.001**
eGFR, mL/min/1.73 m^2^	111 ± 21	107 ± 16	**<0.001**
ACR, mg/mmol crea	0.4 (0.2; 1.0)	0.5 (0.2; 1.3)	**<0.001**
C-reactive protein, mg/L	0.5 (0.2; 1.4)	0.5 (0.2; 2.0)	**0.001**
ADMA, (µmol/L)	0.47 (0.36; 0.60)	0.44 (0.34; 0.57)	**<0.001**
Erythrocytes, 10^12^/L	5.14 ± 0.31	4.55 ± 0.29	**<0.001**
Leukocytes, 10^9^/L	6.35 ± 1.44	6.85 ± 1.78	**<0.001**
**Prevalence**			** *p* **
Elevated:			
Uric acid, *n* (%)	401 (30.9)	81 (5.8)	**<0.001**
Homocysteine, *n* (%)	391 (30.1)	189 (13.5)	**<0.001**
Waist/height, *n* (%)	175 (13.5)	138 (9.8)	**0.004**
Body mass index, *n* (%)	369 (28.4)	261 (18.6)	**<0.001**
Systolic blood pressure, *n* (%)	329 (25.3)	20 (1.4)	**<0.001**
Diastolic blood pressure, *n* (%)	87 (6.7)	56 (4.0)	**0.002**
Blood pressure, *n* (%)	347 (26.7)	65 (4.6)	**<0.001**
Glucose, *n* (%)	84 (6.5)	29 (2.1)	**<0.001**
Insulin, *n* (%)	104 (8.0)	89 (6.3)	0.100
Triacylglycerols, *n* (%)	66 (5.1)	73 (5.2)	0.931
Atherogenic index, *n* (%)	111 (8.6)	55 (3.9)	**<0.001**
C-reactive protein, *n* (%)	98 (7.6)	156 (11.1)	**0.002**
ACR, *n* (%)	48 (3.7)	45 (3.2)	0.527
Low HDL-C, *n* (%)	193 (14.9)	273 (19.5)	**0.001**
MetSy, *n* (%)	53 (4.1)	19 (1.4)	**<0.001**

SBP, systolic blood pressure, DBP, diastolic blood pressure; QUICKI, quantitative insulin sensitivity check index; HDL-C, high-density lipoprotein cholesterol; TAG, triacylglycerols; cMSS, continuous metabolic syndrome score; eGFR, estimated glomerular filtration rate; ACR, urinary albumin-to-creatinine ratio; ADMA, asymmetric dimethylarginine; data are given as mean ± standard deviation, those analyzed after log transformation as back-transformed geometric mean (interval −1 SD, +1 SD), categorical data as counts and frequencies; groups were compared using the two-sided Student’s *t*-test for independent samples or the Fisher’s exact test, *p* < 0.05 is given in bold.

**Table 2 ijerph-19-13521-t002:** Pearson correlations between selected cardiometabolic markers and uric acid or homocysteine in males and females.

Males											
		Age	BMI	WHtR	SBP	DBP	Glucose	Insulin	QUICKI	HDL-C	Non-HDL-C
**UA**	r	−0.029	0.316	0.286	0.152	0.120	0.009	0.121	−0.103	−0.156	0.176
	*p*	0.298	**<0.001**	**<0.001**	**<0.001**	**<0.001**	0.752	**<0.001**	**<0.001**	**<0.001**	**<0.001**
**LnHcy**	r	0.148	0.004	−0.019	0.021	0.073	−0.023	−0.001	0.005	−0.039	0.065
	*p*	**<0.001**	0.889	0.497	0.440	**0.008**	0.401	0.967	0.856	0.156	**0.019**
		**LnTAG**	**AIP**	**cMSS**	**RF No.**	**eGFR**	**LnACR**	**LnADMA**	**LnCRP**	**Ery**	**Leu**
**UA**	r	0.155	0.188	0.270	0.257	−0.062	−0.093	0.063	0.185	0.110	0.067
	*p*	**<0.001**	**<0.001**	**<0.001**	**<0.001**	**0.025**	**0.001**	**0.023**	**<0.001**	**<0.001**	**0.017**
**LnHcy**	r	0.094	0.095	0.051	−0.008	−0.076	−0.056	−0.072	−0.018	0.046	0.051
	*p*	**<0.001**	**0.001**	0.069	0.786	**0.006**	**0.044**	**0.010**	0.521	0.059	0.068
**Females**											
		**Age**	**BMI**	**WHtR**	**SBP**	**DBP**	**Glucose**	**Insulin**	**QUICKI**	**HDL-C**	**Non-HDL-C**
**UA**	r	−0.087	0.253	0.198	0.111	0.110	0.037	0.022	−0.012	−0.134	0.055
	*p*	**0.001**	**<0.001**	**<0.001**	**<0.001**	**<0.001**	0.165	0.410	0.663	**<0.001**	**0.041**
**LnHcy**	r	0.094	0.028	−0.026	0.047	0.053	−0.008	0.012	−0.007	−0.030	0.051
	*p*	**<0.001**	0.303	0.400	0.077	**0.047**	0.756	0.653	0.799	0.258	0.058
		**LnTAG**	**AIP**	**cMSS**	**RF No.**	**eGFR**	**LnACR**	**LnADMA**	**LnCRP**	**Ery**	**Leu**
**UA**	r	0.023	0.081	0.257	0.181	−0.152	−0.043	0.113	0.111	0.126	0.107
	*p*	0.388	**0.002**	**<0.001**	**<0.001**	**<0.001**	**0.023**	**<0.001**	**<0.001**	**<0.001**	**<0.001**
**LnHcy**	r	0.078	0.059	−0.008	0.030	−0.131	0.018	−0.071	0.010	0.053	0.052
	*p*	**0.003**	**0.026**	0.786	0.267	**<0.001**	0.527	**0.009**	0.706	**0.048**	0.053

UA, uric acid; ln, logarithm; Hcy, homocysteine; BMI, body mass index; WHtR, weight-to-height ratio; SBP, systolic blood pressure, DBP, diastolic blood pressure; QUICKI, quantitative insulin sensitivity check index; HDL-C, high-density lipoprotein cholesterol; TAG, triacylglycerols; AIP, atherogenic index of plasma; cMSS, continuous metabolic syndrome score; RF No., number of cardiometabolic risk factors; eGFR, estimated glomerular filtration rate; ACR, urinary albumin-to-creatinine ratio; ADMA, asymmetric dimethylarginine; CRP, C-reactive protein; Ery, erythrocytes; Leu, leukocytes; *p* < 0.05 is given in bold.

**Table 3 ijerph-19-13521-t003:** Characteristics of males according to uricemia and homocysteinemia.

	Normouricemia, *n* = 897 (69.1%)		Hyperuricemia, *n* = 401 (30.9%)		*p*		
	NHcy (71.3%)	HHcy (28.7%)	NHcy (58.6%)	HHcy (41.4%)	UA	Hcy	Interaction
Number	640 (49.3%)	257 (19.8%)	235 (18.1%)	166 (12.8%)	--	--	--
Age, years	17.6 ± 1.4	16.9 ± 1.4	16.8 ± 1.3	16.4 ± 1.1	--	--	--
Uric acid, µmol/L	326 ± 43	325 ± 39	422 ± 42	413 ±40	--	--	--
Homocysteine, µmol/L	9.6 (7.6, 12.1)	16.5 (11.2, 24.2)	9.1 (7.3, 11.4)	14.7 (10.5, 20.5)	--	--	--
Body mass index, kg/m^2^	22.7 ± 3.6	21.9 ± 3.0	24.7 ± 4.5	23.7 ± 4.3	**<0.001**	**<0.001**	0.767
WHtR	0.44 ± 0.05	0.43 ± 0.04	0.47 ± 0.06	0.45 ± 0.06	**<0.001**	**<0.001**	0.406
SBP, mmHg	122 ± 12	120 ± 11	125 ± 12	123 ± 13	**0.001**	**0.008**	0.852
DBP, mmHg	72 ±8	72 ± 7	74 ± 8	73 ± 10	**0.017**	0.179	0.856
Glucose, mmol/L	4.9 ± 0.4	4.9 ± 0.4	4.9 ± 0.4	5.0 ± 0.4	0.376	0.124	0.448
Insulin, μIU/mL	9.1 (5.5, 15.0)	9.5 (5.8, 15.6)	10.4 (5.9, 18.3)	10.3 (6.1, 17.6)	**0.001**	0.504	0.416
QUICKI	0.346 ± 0.027	0.343 ± 0.028	0.339 ± 0.030	0.339 ± 0.028	**0.002**	0.272	0.554
HDL-C, mmol/L	1.27 ± 0.22	1.28 ± 0.24	1.21 ± 0.23	1.19 ± 0.24	**<0.001**	0.839	0.420
Non-HDL-C, mmol/L	2.55 ± 0.66	2.47 ± 0.60	2.64 ± 0.73	2.67 ± 0.75	**0.001**	0.588	0.181
TAG, mmol/L	0.77 (0.51, 1.16)	0.78 (0.54, 1.14)	0.82 (0.50, 1.33)	0.82 (0.52, 1.31)	**0.034**	0.543	0.852
Atherogenic index	−0.21 ± 0.21	−0.21 ± 0.20	−0.16 ± 0.27	−0.15 ± 0.24	**<0.001**	0.537	0.916
cMSS	1.95 ± 0.43	1.91 ± 0.38	2.14 ± 0.59	2.11 ± 0.57	**<0.001**	0.239	0.871
Risk factors number	0.85 ± 0.98	0.72 ± 0.89	1.34 ± 1.18	1.16 ± 1.16	**<0.001**	**0.017**	0.686
eGFR, mL/min/1.73 m^2^	111 ± 22	109 ± 20	113 ± 23	110 ± 20	0.354	0. 085	0.600
ACR, mg/mmol	0.4 (0.2, 0.9)	0.4 (0.2, 1.0)	0.4 (0.2, 1.0)	0.4 (0.1, 0.9)	0.309	0.531	0.102
ADMA, µmol/L	0.43 (0.33, 0.56)	0.47 (0.37, 0.60)	0.50 (0.39, 0.65)	0.52 (0.42, 0.65)	**<0.001**	**<0.001**	0.108
C-reactive protein, mg/L	0.5 (0.1, 1.5)	0.4 (0.1, 1.2)	0.7 (0.2, 2.3)	0.5 (0.2, 1.7)	**<0.001**	**0.001**	0.744
Erythrocytes, 10^12^/L	5.11± 0.30	5.16 ± 0.31	5.19 ± 0.30	5.17± 0.29	**0.021**	0.319	0.071
Leukocytes, 10^9^/L	6.32 ± 1.35	6.34 ± 1.69	6.36 ± 1.52	6.50 ± 1.24	0.268	0.364	0.509

NHcy, normohomocysteinemia; HHcy, hyperhomocysteinemia; UA, uric acid; Hcy, homocysteine; WHtR, weight-to-height ratio; SBP, systolic blood pressure, DBP, diastolic blood pressure; QUICKI, quantitative insulin sensitivity check index; HDL-C, high-density lipoprotein cholesterol; TAG, triacylglycerols; cMSS, continuous metabolic syndrome score; eGFR, estimated glomerular filtration rate; ACR, urinary albumin-to-creatinine ratio; ADMA, asymmetric dimethylarginine; data are given as mean ± standard deviation, those analyzed after log transformation as back-transformed geometric mean (interval −1 SD, +1 SD), groups were compared using the two-factor Analysis of variance (ANOVA), *p* < 0.05 is given in bold.

**Table 4 ijerph-19-13521-t004:** Multivariate regression of uric acid, homocysteine, and cardiometabolic risk factors and markers on continuous metabolic syndrome score using the orthogonal projections to latent structures model in males and females.

	Males				Females			
	Model 1	Model 2	Model 3	Model 4	Model 1	Model 2	Model 3	Model 4
Insulin	**1.93**	**1.89**	**2.06**	**2.01**	**1.57**	**1.63**	**1.66**	**1.71**
lnCRP	**1.19**	**1.21**	**1.24**	**1.25**	**1.50**	**1.57**	**1.58**	**1.65**
Leukocytes	**1.17**	**1.07**	**1.24**	**1.13**	**1.04**	**1.12**	**1.10**	**1.17**
Erythrocytes	0.98	0.97	**1.04**	**1.02**	0.22	0.32	0.24	0.34
lnACR	0.54	0.56	0.62	0.60	0.12	0.15	0.13	0.15
Age	0.44	0.54	0.41	0.48	0.88	0.87	0.92	0.90
lnADMA	0.15	0.33	0.10	0.25	**1.50**	**1.00**	**1.14**	**1.01**
eGFR	0.08	0.17	0.04	0.24	0.40	0.59	0.47	0.66
Uric acid	--	**1.07**	--	**1.13**	--	0.61	--	0.64
lnHcy		--	0.29	0.35	--	--	0.31	0.43
R^2^	32%	35%	32%	35%	22%	23%	22%	22%

Variables with variable of importance for the projection values ≥1.00 were considered as important contributors (given in bold); ln, logarithm; CRP, C-reactive protein; ACR, urinary albumin-to-creatinine ratio; ADMA, asymmetric dimethylarginine; eGFR, estimated glomerular filtration rate; Hcy, homocysteine.

**Table 5 ijerph-19-13521-t005:** Characteristics of females according to uricemia and homocysteinemia.

	Normouricemia, *n* = 1321 (94.2%)		Hyperuricemia, *n* = 81 (5.8%)		*p*		
	NHcy (85.3%)	HHcy (14.7%)	NHcy (81.5%)	HHcy (18.5%)	UA	Hcy	Interaction
Number	1127 (80.3%)	194 (13.8%)	66 (4.7%)	15 (1.1%)	--	--	--
Age, years	17.4 ± 1.4	16.5 ± 1.3	16.9 ± 1.5	16.2 ± 1.3	--	--	--
Uric acid, µmol/L	250 ± 44	263 ± 42	368 ± 35	382 ±45	--	--	--
Homocysteine, µmol/L	8.8 (6.9, 11.1)	14.3 (11.3, 18.1)	9.0 (7.3, 11.1)	14.0 (10.2, 19.4)	--	--	--
Body mass index, kg/m^2^	21.8 ± 3.3	21.7 ± 3.3	24.5 ± 5.2	24.4 ± 4.8	<0.001	0.848	0.990
WHtR	0.43 ± 0.05	0.43 ± 0.04	0.46 ± 0.07	0.46 ± 0.06	<0.001	0.372	0.966
SBP, mmHg	107 ± 9	107 ± 10	112 ± 9	110 ± 6	0.004	0.391	0.446
DBP, mmHg	70 ± 7	70 ± 8	74 ± 7	72 ± 6	0.004	0.262	0.411
Glucose, mmol/L	4.7 ± 0.4	4.7 ± 0.4	4.7 ± 0.4	4.8 ± 0.5	0.367	0.519	0.985
Insulin, μIU/mL	9.9 (6.2, 15.8)	10.1 (6.5, 15.6)	10.8 (5.8, 20.0)	12.0 (7.0, 20.7)	0.065	0.374	0.527
QUICKI	0.344 ± 0.025	0.342 ± 0.024	0.340 ± 0.033	0.333 ± 0.028	0.108	0.273	0.527
HDL-C, mmol/L	1.53 ± 0.30	1.48 ± 0.32	1.47 ± 0.30	1.35 ± 0.24	0.030	0.076	0.415
Non-HDL-C, mmol/L	2.73 ± 0.69	2.74 ± 0.70	2.88 ± 0.64	2.68 ± 0.69	0.637	0.359	0.304
TAG, mmol/L	0.81 (0.53, 1.22)	0.81 (0.55, 1.20)	0.86 (0.56, 1.31)	0.90 (0.60, 1.36)	0.184	0.621	0.660
Atherogenic index	−0.27 ± 0.20	−0.25 ± 0.20	−0.23 ± 0.22	−0.17 ± 0.19	0.035	0.215	0.468
cMSS	1.82 ± 0.40	1.80 ± 0.44	1.97 ± 0.55	2.05± 0.42	0.001	0.697	0.427
Risk factors number	0.64 ± 0.85	0.62 ± 0.88	1.17 ± 1.24	1.33± 1.23	<0.001	0.570	0.480
eGFR, mL/min/1.73 m^2^	108 ± 16	106 ± 15	103 ± 15	100 ± 23	0.028	0. 380	0.889
ACR, mg/mmol	0.5 (0.2, 1.3)	0.5 (0.2, 1.5)	0.4 (0.2, 1.2)	0.6 (0.1, 4.3)	0.896	0.170	0.156
ADMA, µmol/L	0.43 (0.34, 0.56)	0.47 (0.38, 0.59)	0.46 (0.36, 0.60)	0.49 (0.42, 0.56)	0.280	0.111	0.693
C-reactive protein, mg/L	0.5 (0.1, 2.0)	0.4 (0.1, 1.4)	1.0 (0.3, 4.0)	1.4 (0.5, 3.7)	<0.001	0.875	0.149
Erythrocytes, 10^12^/L	4.54± 0.28	4.59 ± 0.32	4.62 ± 0.34	4.60± 0.25	0.328	0.733	0.370
Leukocytes, 10^9^/L	6.82 ± 1.76	6.79 ± 1.74	7.25 ± 1.81	7.73 ± 2.89	0.009	0.392	0.327

NHcy, normohomocysteinemia; HHcy, hyperhomocysteinemia; UA, uric acid; Hcy, homocysteine; WHtR, weight-to-height ratio; SBP, systolic blood pressure, DBP, diastolic blood pressure; QUICKI, quantitative insulin sensitivity check index; HDL-C, high-density lipoprotein cholesterol; TAG, triacylglycerols; cMSS, continuous metabolic syndrome score; eGFR, estimated glomerular filtration rate; ACR, urinary albumin-to-creatinine ratio; ADMA, asymmetric dimethylarginine; data are given as mean ± standard deviation, those analyzed after log transformation as back-transformed geometric mean (interval −1 SD, +1 SD), groups were compared using the two-factor Analysis of variance (ANOVA).

## Data Availability

The data that support the findings of this study are available from the corresponding author upon reasonable request.

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
