# Peer review of "The Presence of Hyperhomocysteinemia Does Not Aggravate the Cardiometabolic Risk Imposed by Hyperuricemia in Young Individuals: A Retrospective Analysis of a Cross-Sectional Study"

_ijerph, 2022, doi:10.3390/ijerph192013521_

Round 1

Reviewer 1 Report

This study is interesting and generally well written. A few minor edits have been suggested. However, one major concern relates to the exclusion criteria that was applied after data collection.

It is unclear why non-Caucasians were excluded from data analyses. This seems unethical and the data should be included. It is also relevant to include the number of participants excluded for each of the listed categories (how many of the 260 were non-Caucasian, potential diabetics, etc.) How was ‘potential diabetic’ determined?

Minor comments:

Fix font in lines 200-202

Remove highlighting line 271

Author Response

We are thankful to the reviewer for her/his time spent and for their suggestions on how to improve our paper. In the revised version of the manuscript, we tried to incorporate the issues raised. The implemented changes are highlighted in the revised paper. 

Reviewer 1

This study is interesting and generally well written. A few minor edits have been suggested. However, one major concern relates to the exclusion criteria that was applied after data collection.

It is unclear why non-Caucasians were excluded from data analyses. This seems unethical and the data should be included. It is also relevant to include the number of participants excluded for each of the listed categories (how many of the 260 were non-Caucasian, potential diabetics, etc.) How was ‘potential diabetic’ determined?

As suggested by the reviewer, we indicate the number of excluded subjects. e.g., non-Caucasians: 9, potential diabetics: 6 (FPG>6.9mmol/L), 11 younger than 14 years and 14 older than 20 years, missing data: 220, in the revised version of the manuscript (lines:90-93).

In contrast to the reviewer, we consider the exclusion of the non-Caucasian students (representing 0.003% of the evaluated cohort) rationale and substantiated, and their exclusion should in no way be interpreted as discrimination of minorities. In the “Respect for Health” survey, every student who wished, and whose parents agreed, could participate. A few non-Caucasian students were excluded from the current evaluation, as owing to the small numbers of non-Caucasian residents in Slovakia we have no established reference ranges (e.g., growth curves, blood chemistry variables, etc.) for non-Caucasian minorities. A small group of non-Caucasian students could not be evaluated per se. If included, it could have raised similar questions as their exclusion did.

As to the small numbers of non-Caucasian residents in Slovakia: our study has been conducted in the 2011/2012 school year. According to data from 2012, out of 5 410 836 inhabitants of Slovakia, 67 635 (1.25%) were foreigners, among them 37.5% with origin out of EU or EEC countries. Vietnamese accounted for about 3% of foreigners, proportions of other minorities represented in our study are not detailed in statistical reports as to their very small frequencies.

Minor comments:

Fix font in lines 200-202

We apologize for this mistake, it was corrected in the revised paper.

Remove highlighting line 271

We apologize for this mistake, it was corrected in the revised paper

Reviewer 2 Report

Little research was conducted into the effects of the combined manifestation of hyperuricemia and hyperhomocysteinemia on cardiometabolic risk factors and markers in young subjects. 1,298 males and 1,402 females, 14-to-20-year-olds, were classified into four groups, as presenting or not either hyperuricemia or hyperhomocysteinemia. Anthropometric measures, blood pressure, plasma glucose, insulin, lipids, markers of renal function, C-reactive protein, asymmetric dimethylarginine, and blood counts were determined. Hyperuricemic males (but not females) had higher odds for hyperhomocysteinemia than normouricemic ones (OR: 1.8; 95% CI: 1.4-2.3; p Among four groups, subjects concurrently manifesting hyperuricemia and hyperhomocysteinemia did not presented the highest continuous metabolic syndrome score – a proxy measure of cardiometabolic risk; neither the multivariate regression model indicated a concurrent significant effect of uric acid and homocysteine on continuous metabolic syndrome score in either sex. The authors concluded that in young healthy subjects, hyperhomocysteinemia does not aggravate negative health effects imposed by hyperuricemia. While this work is interesting, a number of concerns remain.

1.     This study generated a negative finding in young populations and thus it is intriguing to know if this is the case for older populations. Older age seems to be a necessary control group.

2.     There are many language mistakes. For example, “….. were classified into four groups, as presenting or not either hyperuricemia or hyperhomocysteinemia” contains awkward writing style. Use either “the kidney” or “kidneys” rather than “the kidneys”.

3.     Quality of data needs major improvement.

Author Response

We are thankful to the reviewer for her/his time spent and for their suggestions on how to improve our paper. In the revised version of the manuscript, we tried to incorporate the issues raised. The implemented changes are highlighted in the revised paper. 

Reviewer 2

Comments and Suggestions for Authors

Little research was conducted into the effects of the combined manifestation of hyperuricemia and hyperhomocysteinemia on cardiometabolic risk factors and markers in young subjects. 1,298 males and 1,402 females, 14-to-20-year-olds, were classified into four groups, as presenting or not either hyperuricemia or hyperhomocysteinemia. Anthropometric measures, blood pressure, plasma glucose, insulin, lipids, markers of renal function, C-reactive protein, asymmetric dimethylarginine, and blood counts were determined. Hyperuricemic males (but not females) had higher odds for hyperhomocysteinemia than normouricemic ones (OR: 1.8; 95% CI: 1.4-2.3; p Among four groups, subjects concurrently manifesting hyperuricemia and hyperhomocysteinemia did not presented the highest continuous metabolic syndrome score – a proxy measure of cardiometabolic risk; neither the multivariate regression model indicated a concurrent significant effect of uric acid and homocysteine on continuous metabolic syndrome score in either sex. The authors concluded that in young healthy subjects, hyperhomocysteinemia does not aggravate negative health effects imposed by hyperuricemia. While this work is interesting, a number of concerns remain.

  1. This study generated a negative finding in young populations and thus it is intriguing to know if this is the case for older populations. Older age seems to be a necessary control group.

We are aware of the fact that our cross-sectional analyses generated negative results and we suppose that it is important to communicate them. Unfortunately, we have no follow-up data, which could shed a light on whether hyperhomocysteinemia would aggravate the negative effects of hyperuricemia in the later life of our students. As our paper is submitted to a special issue of IJERPH entitled 2nd Edition of Adolescent and Young People's Health Issues and Challenges”, we suppose that the inclusion of an additional independent cohort of adults is beyond the scope of this special issue. In limitations, we mention that our data cannot be generalized to other populations, which, to our perception, includes also older populations. In the conclusions, we state that our data do not rule out that there are (even non-clinical) populations where the combined hyperuricemia and hyperhomocysteinemia are associated with an increased cardiometabolic risk. We expect (and even hope) that our negative data could be a challenge for other groups to study the effects of concurrent manifestation of hyperuricemia and hyperhomocysteinemia in different epidemiological or clinical cohorts.

  1. There are many language mistakes. For example, “….. were classified into four groups, as presenting or not either hyperuricemia or hyperhomocysteinemia” contains awkward writing style. Use either “the kidney” or “kidneys” rather than “the kidneys”.

We are thankful to the reviewer for pointing out these mistakes. As suggested, we corrected these mistakes in the revised paper (lines 17-19, 59).

  1. Quality of data needs major improvement.

We apologize but we do not understand what the reviewer meant by this comment - thus, how should the quality of presented data be improved. To our opinion, the presentation of data fulfills the requirements for presenting data on retrospective analysis of a cross-sectional study.

Round 2

Reviewer 1 Report

Thank you for the revisions.